# High diversity of fungal ecological groups in Andean–Patagonian *Nothofagus* forests

**Max Emil Schön**[1]*, **Sarah Zuern**[2], **Milena Vera**[2], **Sigisfredo Garnica**[2]*

**1** Department of Biomolecular Mechanisms, Max Planck Institute for Medical Research, Heidelberg, Germany, **2** Instituto de Bioquímica y Microbiología, Facultad de Ciencias, U Austral de Chile, Isla Teja, Valdivia, Chile

* max-emil.schoen@mr.mpg.de (MES); sigisfredo.garnica@uach.cl (SG)

**Data Availability Statement:** Raw sequencing data files are available from the Sequence Read Archive (SRA) under BioProject number PRJNA970541 (http://www.ncbi.nlm.nih.gov/bioproject/970541).

## Abstract

Native Andean–Patagonian *Nothofagus* forests harbour a unique diversity of microorganisms with diverse ecological roles. Although ectomycorrhizal associations constitute an important fragment of the biota associated with these forests, the factors affecting such communities are largely unknown. We investigated the biodiversity, relative abundance, and composition of ectomycorrhizal fungal communities in relation to two host tree species and ages and the soil properties in six monospecific and mixed evergreen–deciduous *Nothofagus* forests. We used the internal transcribed spacer (ITS2) region by sequencing 9,600 ectomycorrhizae (ECM) root tips for the identification of fungi. In total, 1,125 fungal taxa at the genus level distributed over 131 orders were identified. The phyla Ascomycota (34.5%) and Basidiomycota (62.1%) were the most abundant, whereas Mucoromycota (3.1%), Chytridiomycota, Cryptomycota, Olpidiomycota, and Zoopagomycota occurred less frequently. The highest taxon diversity was found in old trees, whereas young trees often exhibited a lower diversity of the associated fungi. The fungal taxa were grouped into seven broad ecological categories, of which saprotrophic associations were most common, followed by pathotrophic, pathotrophic–saprotrophic–symbiotrophic, pathotrophic–saprotrophic, and symbiotrophic associations. We did not detect significant differences in the number of taxa in each category between young and old *N. dombeyi* and *N. obliqua*. Overall, the scale of the Illumina sequencing approach allowed us to detect a fungal taxa diversity that would not be possible to find through surveys of fruiting bodies alone and that have never been observed in *Nothofagus* forests before. Our findings suggest the impact of the proximity between sites, the similarity of the soil conditions, and anthropogenic use of the forests on the belowground fungal community's diversity and composition. Furthermore, there were differences between above- and belowground occurrences of the edible mushrooms *B. loyo* and *Ramaria* spp. However, future research, including on EMC tips found beneath fairy rings could provide significantly better correlations with the occurrence of aboveground fruiting body.

## Introduction

Fungi play pivotal ecological roles in forest ecosystems and some of them represent an important forest resource [1, 2]. They may participate in the decomposition of wood and litter

**Funding:** SG: The field and laboratory research were financially supported by the Fundación para la Innovación Agraria (FIA) belonging to the Ministry of Agriculture of the Government of Chile (Grant FYT-2018-0723). MES: This work was supported by the Max Planck Society. The funders had no role in study design, data collection and analysis, decision to publish, or preparation of the manuscript.

**Competing interests:** The authors have declared that no competing interests exist.

recycling of carbon, minerals, and nutrients that other organisms can use [3]. Alternatively, they can act as pathogens, killing trees, thus altering plant diversity, or collaborating symbiotically with plants and promoting forest growth [4].

Research on fungi in forest ecosystems has comes mostly from boreal and temperate forests, especially in North America and Europe [5, 6]. In contrast, our knowledge of these microorganisms in other regions, such as the temperate forests of South America, is still very fragmentary. This applies both to their aboveground fruiting bodies and to their underground life, which have mostly been studied by root tip sequencing (as reviewed by Barroetaveña et al. [7]). Recently, a high diversity of fungi, far beyond that detected from fruiting bodies, has been revealed to be associated with *Nothofagus* species [8–10]. Members of this genus, also known as the southern beeches, constitute the main part of forests in the southernmost parts of Chile and Argentina, and also in parts of Australia, New Zealand, and Papua New Guinea. Understanding the fungal partners of these trees with also help to elucidate factors impacting these less well studied forest ecosystems.

An important functional group of wild mushrooms are those that establish obligatory symbiotic associations with the roots of evergreen–deciduous *Nothofagus* species known as ectomycorrhizae (ECM). Nouhra et al. [8] found no significant differences between different *Nothofagus* species linked to the diversity and composition of ECM fungal communities, but altitude affected their structure. Truong et al. [9] found similar functional fungal groups across elevations in *N. pumilio* forests in Chile and Argentina, and Almonacid-Muñoz et al. [10] found that microbial communities in *N. obliqua* are affected by the surrounding forest and the site's characteristics.

In Chile, fungi growing in native forests are considered non-timber forest products [11]. Some wild mushroom species are also important as food, especially for indigenous Mapuche communities [2, 12]. Despite their ecological importance and some of them being a source of food [13], their occurrence, as evaluated through the production of fruiting bodies, is apparently decreasing through anthropogenic pressure, especially the loss, alteration, and/or fragmentation of their habitats [14]. Thus, research aimed at expanding knowledge about the biotic and abiotic factors that influence wild mushroom communities is vital for their management and conservation.

In this work, temperate rainforests were mostly sampled in the commune of Panguipulli, which is recognized by UNESCO as a Biosphere Reserve [15]; it is a commune with the largest area of native forest and the largest Mapuche population in the Los Ríos region of Chile. The native forest covers more than 95%, and *Nothofagus*, in turn, covers 66.7% of the communal surface. The two species *N. dombeyi* (evergreen) and *N. obliqua* (deciduous) are the dominant representatives [16]. In this commune, around 42% of the population are members of the Mapuche people [17]. This aboriginal population still preserves many elements of their identity and culture, including knowledge of the natural environment [15, 18]. In particular, gathering wild mushrooms in these forests is a cultural practice that has disappeared rapidly in recent years. This is partly because the transmission of this tradition is declining but also because of the extinction of traditional mushrooms such as the ecomycorrhizal "loyo" (*Boletus loyo* Phil.) and "changles" (*Ramaria* spp.). These species often fructify in fairy rings associated with trees of the genus *Nothofagus*, specifically the evergreen *N. dombeyi* and the deciduous *N. obliqua*. Both *B. loyo* and *Ramaria* spp. are important sources of food, but have become less frequent or have already disappeared entirely, especially in sites with strong anthropogenic interventions, such as logging [19]. Therefore, the Chilean Ministry of the Environment recently declared the loyo to be an endangered species. In addition, it should be noted that Panguipulli is the commune with a notable rapprochement between collectors of wild edible mushrooms and researchers. This has led to several workshops for

the exchange of knowledge, thus creating a record of wild mushrooms known by collectors in the territory [20].

This article presents the findings from a research project carried out in southern Chile (Fig 1A). In the framework of the project, the diversity of ECM fungi associated with *Nothofagus* spp. from fruiting bodies and ECM tips was studied. The aim was to develop methods of propagating edible wild mushrooms and inoculating native forest plots with them in order to promote sustainable mushroom collection as an alternative for the diversification of family farming [21]. As part of this project, we have been conducting research and collecting fruiting bodies for several years, specifically targeting edible ECM mushrooms from these forests.

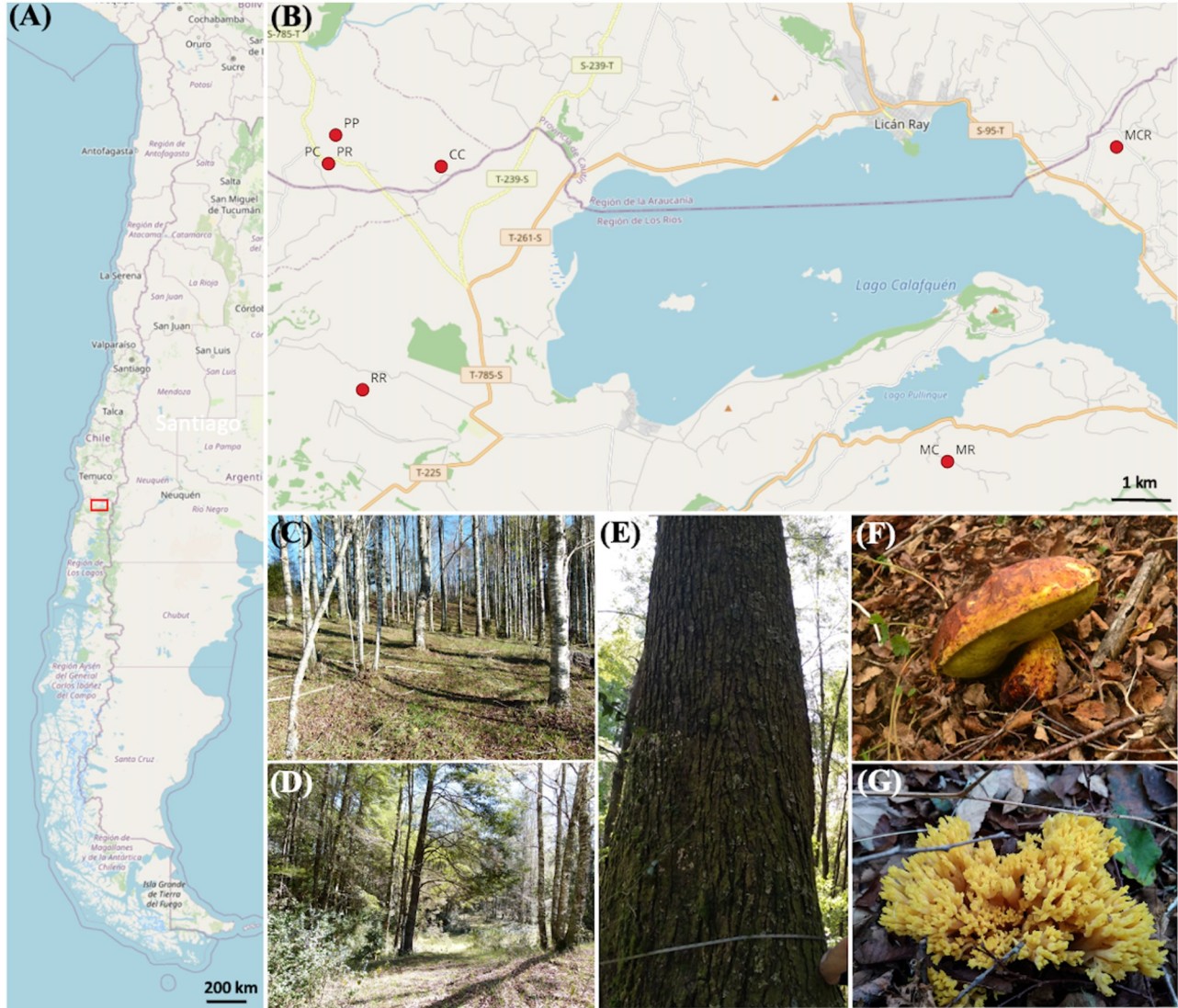

**Fig 1. Geographic location, sampling sites, and edible fruiting bodies in the communes of Loncoche in La Araucania and Panguipulli in the Los Ríos region (Chile).** (A) Map of Chile, indicating the geographic location (red box) where the material was collected. (B) Map representing a magnification of the red box from A, Collection sites in the Loncoche commune: The sites at Pultro road (PR and PC), and (PP) Pultro meadow site. Collection sites in the Panguipulli commune: Caricuicui (RR), Linda Flor (CC), Pullinque Alto (MR and MC), and Pukura (MCR). (C) Young pure stand of *N. obliqua*. (D) Young mixed stand of *N. dombeyi* and *N. obliqua*. (E) Old *N. dombeyi* tree. (F) Mature fruiting body of *Boletus loyo*. (G) Fruiting body of *Ramaria* sp. ©OpenStreetMap contributors (2015); maps retrieved in June 2023 from https://planet.openstreetmap.org.

Here, we provide information about the ECM fungal communities by sampling the mycorrhizal root tips from *N. dombeyi* and *N. obliqua* from both pure and mixed forests, and addressing the following specific questions:

1. What is the diversity of the fungi associated with the roots of *N. dombeyi* and *N. obliqua* from pure and mixed forests?

2. What is the relative belowground abundance of the wild edible mushrooms *B. loyo* and *Ramaria* spp. in the sampled forests?

3. Do biotic and/or abiotic factors have an impact on the fungal communities in general and the edible mushrooms *B. loyo* and *Ramaria* spp. in particular?

We hypothesized that the ECM fungal communities would differ across evergreen–deciduous *Nothofagus* forests represented either by pure or mixed stands of *N. dombeyi* and/or *N. obliqua*. This continuum can be correlated to changes in biotic and/or environmental abiotic variables such as the soil's chemical properties, plant species, and tree age. We aimed to contribute to the knowledge of the diversity and relative abundance of ECM fungal communities associated with the roots of these two *Nothofagus* species. A special focus was to characterize the relative abundance of the edible fungi *B. loyo* and *Ramaria* spp. associated with the roots of two different species of *Nothofagus*. The results generated here are intended to promote actions for these two sources of food and other species of mushrooms in order to apply appropriate sustainable silvicultural management, conservation decisions, and restoration actions.

## Material and methods

### Ethics statement

The plant species used in this study are not protected and therefore no specific permits were requested for sampling. For sampling in the communes of Loncoche and Panguipulli (Chile), we received oral permission from the individual owners, namely Mauricio Ibarra (Pultro), Rosario Catripán (Caricuicui), José Cayulef (Linda Flor), Matusalem Huenchuanca (Pullinque Alto), and María Chincolef (Pukura), to collect plant material during the execution phase of our project.

### Study sites

The study area is located at the northeast of the Los Ríos region, Chile, commonly known as the "Seven Lakes" because of the number of lakes it hosts, bordering the La Araucania region to the north and Argentina to the east. Specifically, most of the sites are located within the commune of Panguipulli, which is the largest commune of the region with an area of 3,292 km$^2$ (Fig 1A). The sites at Pultro Road (PR and PC) and a meadow in Pultro (PP) are located further north within the La Araucania region, whereas the sites at Caricuicui (RR), Linda Flor (CC), Pullinque Alto (MR and MC), and Pukura (MCR) are within the Los Ríos region (Fig 1B). Geologically, the sites are part of the Andean foothills, whose origin corresponds to the Quaternary period characterized by intense volcanic activity and periods of glaciation that gave rise to various forms of morainic, fluvioglacial, and glaciolacustrine deposits. The climate is temperate-rainy with Mediterranean influences, and the precipitation is concentrated in the winter months (between May and August), ranging between 567 and 623 mm per month. The average temperature reaches 11.5˚C, with the warmest months fluctuating between 13˚C and 16˚C and the winter season months reaching temperatures in the order of 7˚C [22].

The forest in the study area is dominated by the evergreen tree species *Nothofagus dombeyi* (Mirb.) Oerst. ("coihue") and the deciduous *N. obliqua* (Mirb.) Oerst., which is known as

"hualle" in the young growth stage or "pellín" in the case of old trees. The distribution patterns of these two species are linked to different climate and soil conditions; *N. dombeyi* is widespread and occurs best on humid slopes of the west side of the Andes on relatively shallow soil, making it a pioneer species in the Andes Cordillera. *N. obliqua* is a thermophilic species that grows on deep soil, especially in the intermediate depression [23, 24]. The forests at PR/PC and MR/MC are mixed stands; the forest at CC is a pure stand of *N. dombeyi*, and forests at PP, RR, and MCR are pure stands of *N. obliqua*. Fruiting of *B. loyo* and *Ramaria* spp. often occurs in fairy rings from early April until the end of May. The fruiting bodies of *B. loyo* were common in the sites PR, MR, MC, and PCR, whereas *Ramaria* spp. was abundant in the sites RR, CC, MCR, and PP (Fig 1F and 1G). All these forests consist of >300-year-old individual trees surrounded by secondary growth of mostly <70-year-old trees [24]. S1 Table details the main plant species forming part of the understorey and soil properties at each sampled forest site.

## Root and soil sampling

During the summer (January) of 2021, root samples were collected from sites either in monospecific forests composed of *N. obliqua* or *N. dombeyi*, or where both species were mixed (Fig 1C–1E). The collection the sites PR and PC were neighbours to each other or even had some degree of overlap between them; the same applies for the sites MR and MC. In these sites, *N. obliqua* and *N. dombeyi* co-occurred, but each species was sampled and analyzed separately. At each site, 12 trees were randomly selected and root samples were obtained from each: six young trees and six old trees for which the diameter at breast height (DBH) was recorded, where those with DBH ≤ 7 cm were considered to be young trees and those with DBH ≥ 43 cm were classed as old trees. Root samples and the soil covering these were collected within fairy rings at 0.2 m, 0.4 m, and 0.6 m from the base of the trunk for young trees, and 0.5 m, 1 m, and 1.5 m for old trees, at a depth of 20 cm following the direction of the ground slope downwards. Sampling roots from both *Nothofagus* species was especially challenging because their roots grow deep into the soil and, therefore, it was not possible to collect ECM tips beneath the fairy rings (beneath the fruiting bodies). In total, 96 samples containing roots and soil were placed in plastic bags, labelled, and stored in a cooler for transport to the Mycology Laboratory of the Universidad Austral de Chile for storage at 4˚C until processing.

## Root processing and morphotyping

The roots of *N. dombeyi* and *N. obliqua* were separated from the soil and washed with distilled water to remove soil particles adhering to their surface. From each sample, the roots were deposited into Petri dishes for examination under a Zeiss stereomicroscope at 10× and 40× magnification. The ECM root tips were then classified primarily based on their morphology, i.e., branching type, size, colour, and the presence of emanating hyphae and rhizomorphs [25]. Two to five ECM root tips were chosen arbitrarily from those dominant morphotypes. From each sample, 100 ECM tips were obtained, placed in a sterile 1.5-mL Eppendorf tube, and air-dried at 50˚C overnight. In total, 9,600 ECM root tips, half each from young and old plants each were sampled for *N. dombeyi* and *N. obliqua*. Additionally, a part of the root system from each sampled plant was preserved in 70% ethanol and refrigerated at 4˚C.

## DNA extraction and PCR amplification

Genomic DNA was extracted from 96 pooled ECM tip samples using the DNAeasy® Power-Soil® Pro kit (QIAGEN, Hilden, Germany), following the manufacturer's recommendations.

First, previously dried roots were incubated for 30 min in the presence of a buffer and macerated at least three times with a sterile pestil. Next, samples were ground to a fine powder by placing the tissues in a 2.0-mL screw-cap tube containing a single 3.0-mm and five to ten 1.5-mm stainless steel beads and shaking them in a BeadBug™ microtube homogenizer (Sigma-Aldrich, Missouri, USA) for 120 seconds at a speed of 400 rpm. The extracted genomic DNA was stored at −20˚C for later use.

To amplify the internal transcribed spacers (ITS region including the 5.8S), we used the primers ITS1F [26] and ITS4 [27]. The PCR profiles and temperature programs were the same as in Schön et al. [28].

## Library preparation and Illumina sequencing

The quality of the amplified DNA was determined by electrophoresis using a 0.8% agarose gel and by DNA quantification using a 133 Nano Spectrophotometer (NanoDrop Technologies Inc., Wilmington, DE). For Illumina sequencing, the PCR products were pooled 2 to 1 within each site and tree age, resulting in 48 pooled PCR products for sequencing on the Illumina platform (S2 Table). The primers ITS1FI2_A (5′-GAACCWGCGGGGARGGATCA-3′) and ITS2_A (5′-GCTGCGTTCTTCATCGATGC-3′) from the work of Schmidt et al. [29] were used to sequence both DNA strands. The DNA pool from the mixture was prepared for sequencing using the library denaturation and dilution guide according to Fadrosh et al. [30]. Pairwise sequencing was performed using the MiSeq Reagent Kit v3 (2 × 300 cycles) on the MiSeq Illumina sequencing platform at the AUSTRAL-omics Core Facility (Faculty of Science, Universidad Austral de Chile; https://australomics.cl/).

## Analysis for the identification of fungal communities

The demultiplexing process was performed from the raw sequences generated by the Illumina MiSeq with Cutadapt 2.10 [31]. Briefly, Cutadapt identifies and separates sequences from each index in R1, then identifies and groups sequences that match the index in R2. This allows the separation of each sample, depending on the combination of indexes used.

A filtering process was carried out on sequences with a quality lower than Q30. For this, two programs, Trimomatic 0.39 [32] and PRINSEQ 0.20.4 [33], were used to remove the sequences from Illumina sequencing, and poor-quality or partial reads. Subsequently, the paired sequences (forward and reverse) were assembled using PANDAseq 2.11 [34]. Finally, all sequences with a length below 150 bp were removed.

We removed potentially chimeric sequences using the 'uchime_ref' option in Vsearch 2.15.1 [35] and the same database that was used for taxonomic annotation. All non-chimeric sequences were then taxonomically identified by the MALT 0.5.3 pipeline [28, 36], using the NCBI's fungal ITS RefSeq database (PRJNA177353, retrieved 7 September 2021). MALT performs semiglobal alignment of reads to full-length reference sequences and assigns taxonomic labels based on the weighted lowest common ancestor algorithm applied to the top 5% of hits. The results from MALT were tabularized using tools provided by MEGAN 6.21.12 [37] and summarized using Ete3 3.1.2 [38].

In order to compare the composition of the dominant elements of the fungal communities at the different sampling sites, the relative abundance of the most abundant fungal orders was visualized as stacked bar charts in Seaborn 0.11.2 [39].

Since the ITS RefSeq project lacks a reference sequence for the common mushroom species *B. loyo*, we additionally compared reads against the available sequence KY462402.1, using the same approach as described above (MALT and MEGAN).

### Ecological characterization of fungal communities

In addition to its taxonomic classification, the lifestyle of a fungal taxon and the interactions of different life forms in a community are important aspects of fungal ecology. Therefore, all taxa (at the level of genus) were classified into ecological guilds with the FUNGuild tool v1.2 and database (the database is available at www.stbates.org/funguild_db.php) [40]. The database currently consists of more than 13,000 records.

The alpha diversity of the fungal communities was assessed by calculating the Hill numbers [41] based on the normalized counts at the genus level in the R package vegan 2.6–4 [42]. As described in Schön et al. [28], Hill numbers are a general measure of biodiversity, defined as the reciprocal mean proportional abundance, and they assign different weights to taxa based on their abundance (i.e., different weighting for rare and abundant species or taxa). Hill numbers can be computed with different values for the order of diversity $q$, where values of $q < 1$ favour rare taxa (and $q = 0$ simply corresponds to taxon richness), and values of $q > 1$ give more weight to common species or taxa. They also encompass other common diversity indices such as the Shannon index ($q = 1$ is equivalent to the exponential of the Shannon index) and the Simpson index ($q = 2$ is equivalent to the inverse of the Simpson index).

Finally, the relationship between the genus composition of the samples (using only genera with at least 0.1% relative abundance) and the measured environmental parameters, such as inclination, sodium content, and clay portion of the soil, was visualized by non-metric multidimensional scaling (NMDS) based on Bray–Curtis distances in the vegan package ($k = 5$) [42]. Environmental variables were fitted using the 'envfit' function of vegan. The significance of the variables was assessed with 999 random permutations and only significant variables ($p < 0.01$) were plotted.

## Results

### Read counts and chimeric proportions vary across sites

For the pooled samples, the numbers of raw reads ranged from 21,622 to 218,311 and, after filtering, varied between 8,813 and 156,496 (S2 Table). The number of assembled reads ranged from 8,101 to 103,199 reads, with a maximum length of 557 bp. The percentage of chimeric sequences ranged from 1.5% to 49.2% (with an average of 9.1%) for assembled sequences with a minimum length of 150 bp.

### Basidiomycota dominate the fungal communities in *Nothofagus dombeyi* and *N. obliqua* forests

In total, 1125 fungal taxa at the genus level were identified (S3 Table, S1 Fig). These were distributed over 131 orders (S4 Table, Fig 2) representing the phyla Ascomycota (34.5% of total non-chimeric reads), Basidiomycota (62.1%), and Mucoromycota (3.1%), whereas Chytridiomycota, Cryptomycota, Olpidiomycota, and Zoopagomycota were each represented by less than 1% of the total reads (Fig 2 and S5 Table, S2 Fig).

At most sites and for both ages of the tree species sampled, taxa of the orders Agaricales, Helotiales, Thelephorales, Cantharellales, Trichosponorales, and Sebacinales were the most abundant (at least 4% mean relative abundance). At the sites CC, MCR, and PP, both old trees of *N. dombeyi* and *N. obliqua* showed a higher relative abundance of some orders, such as Agaricales, than younger trees (Figs 2 and 3). On the contrary, at the sites MC, MR, and PR, young trees of *N. dombeyi* and *N. obliqua* showed a higher relative abundance of Agaricales. For Helotiales, a high abundance with old trees was detected at the sites CC, MC, and MCR, and with young trees at the sites MR, PP, and RR. Thelephorales were abundant at the sites

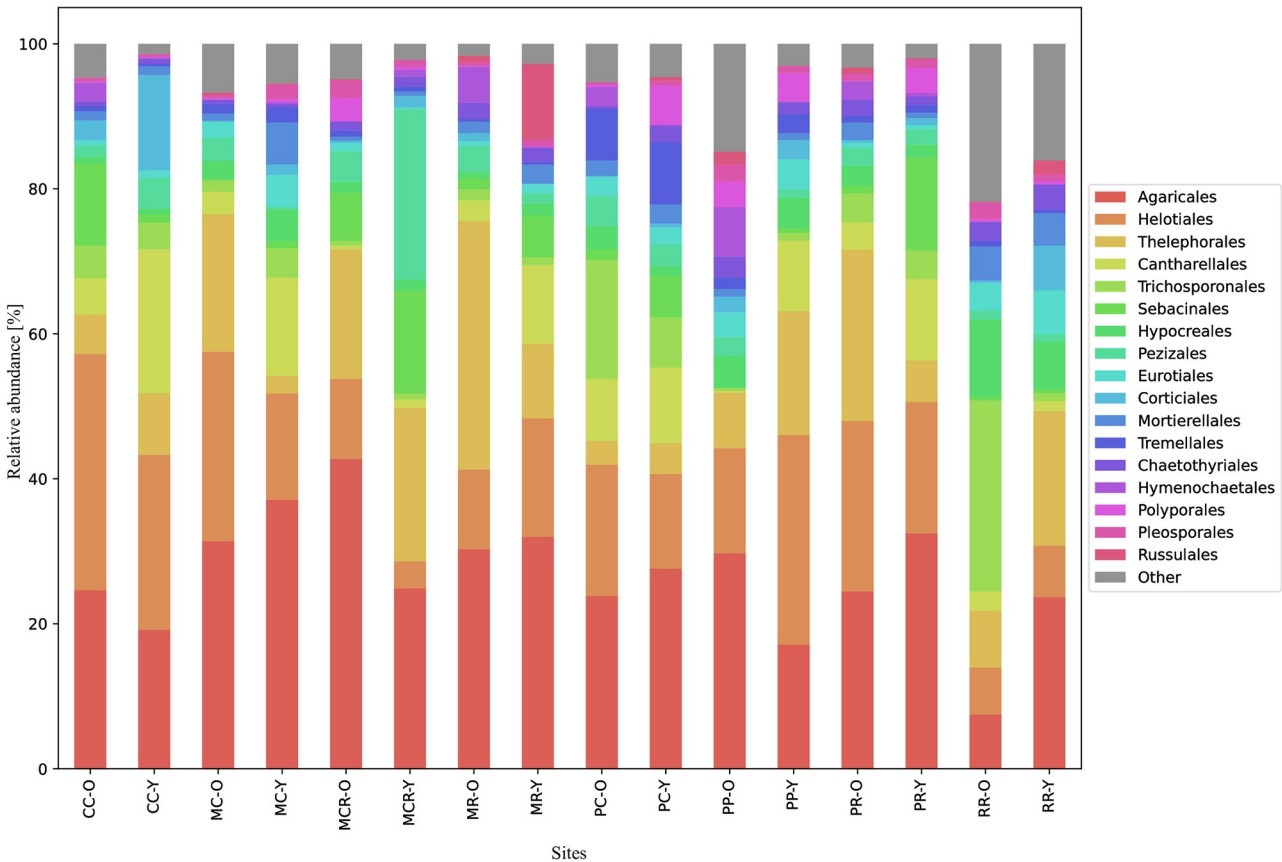

**Fig 2. Relative abundance of the fungal communities at the order level detected in root tips and the surrounding soil from *Nothofagus dombeyi* and *N. obliqua* stands in old (O) and young (Y) trees.** For the site abbreviations, see Fig 1. Fungal orders are ordered according to their mean relative abundances. Orders with a low relative abundance are grouped into 'Other'.

MC, MR, and PR with old trees and at the sites CC, MCR, PC, PP, and RR with young trees. For the case of Cantharellales, in all sites (except the sites CC and RR), the young trees showed a higher abundance. Sebacinales exhibited abundance exclusively in old trees from the sites CC and RR, whereas in all other sites, they were predominantly found in young trees.

At the genus level, the presence of *Boletus* (*B. loyo*) and *Ramaria* (*Ramaria* spp.) was nil or very low in relative abundance at these sites where fruiting bodies of these genera were observed (S3 Table).

The diversity indices of the belowground fungal communities associated with *N. dombeyi* and *N. obliqua* forests are illustrated in S3 Fig (S6 Table). The highest taxon diversity (with $q = 0$, corresponding to species richness) was found in old trees at site RR (589 genera detected), whereas the lowest diversity was found in young trees at site MCR (308 genera). We generally observed a slight trend of lower diversity in young trees vs. old trees (this was easily observable at the sites RR, MCR, and CC).

## Fungal communities do not differ significantly across sites

The composition of the overall fungal communities assessed by two-dimensional NMDS plots based on Bray–Curtis dissimilarities did not show marked differences among the forests/stands sampled (Fig 4; final dimensions, 5; stress, 0.03492074). Moreover, the fungal community composition was not significantly different between young and old trees. Similarly, the

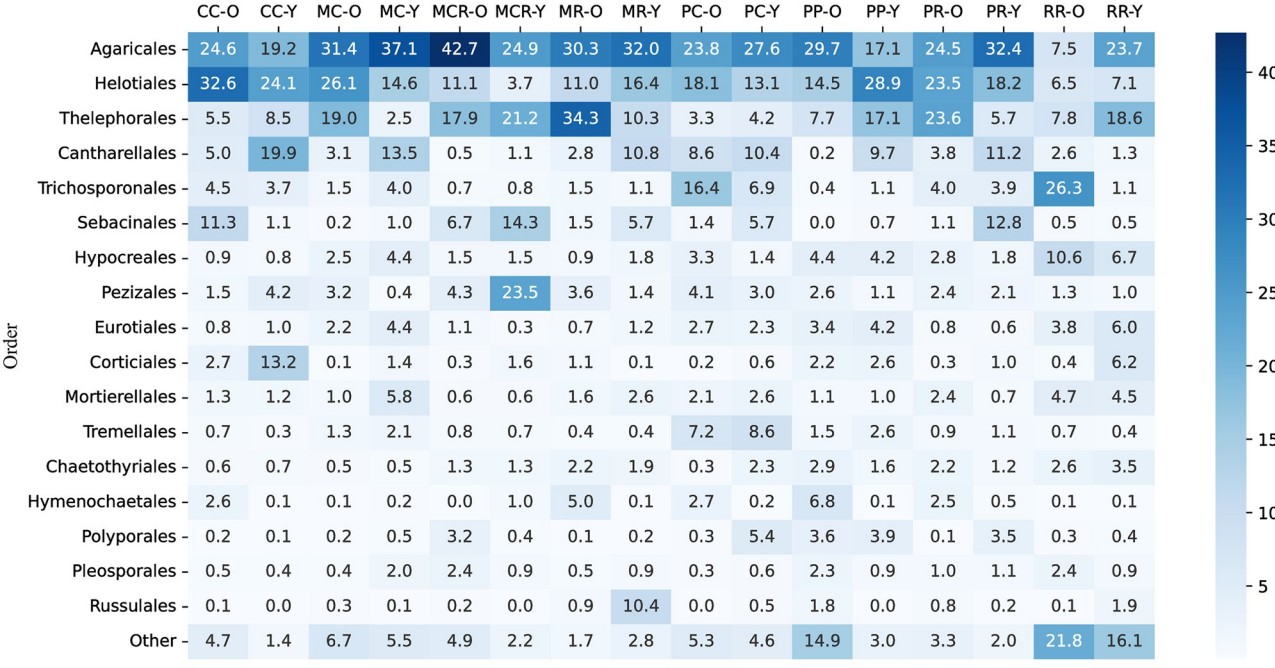

**Fig 3. Heatmap of the fungal communities at the order level detected in root tips and the surrounding soil from *Nothofagus dombeyi* and *N. obliqua* stands in old (O) and young (Y) trees.** For the site abbreviations, see Fig 1. Fungal orders are ordered according to their mean relative abundance. Orders with a low relative abundance are grouped into 'Other'.

soil properties and forest structure (monospecific vs. mixed forests) did not seem to have a significant influence on the fungal community's composition, although non-significant trends could be observed, e.g., along the gradient of total N concentration in the forest soils. The sites could also not be distinguished on the basis of their host trees, but a trend could be observed here as well, with some sites overlapping more than others, especially the sites from mixed stands in the sites MR/MC and PR/PC.

### Saprotrophs are most diverse in fungal communities associated with the root tips (ECMs) and surrounding soil of young and old *Nothofagus dombeyi* and *N. obliqua* trees

The fungal taxa were grouped into seven broad ecological categories; the most common associations were saprotrophic, with a median of 124 taxa; pathotrophic, with 52 taxa, and pathotrophic–saprotrophic–symbiotrophic, pathotrophic–saprotrophic, and symbiotrophic, with a median between 33 and 43 taxa each (Fig 5, S3 Table). There were no significant differences in the number of taxa in each category between young and old *N. dombeyi* and *N. obliqua* trees.

## Discussion

### Diversity of fungal communities in *Nothofagus dombeyi* and *N. obliqua* forests

Our results revealed a high diversity of fungi associated with *N. dombeyi* and *N. obliqua* (see Figs 2 and 3). In general, the fungal groups detected here are in agreement with previous studies conducted in other South American *Nothofagus* forests [8, 9] and even studies conducted

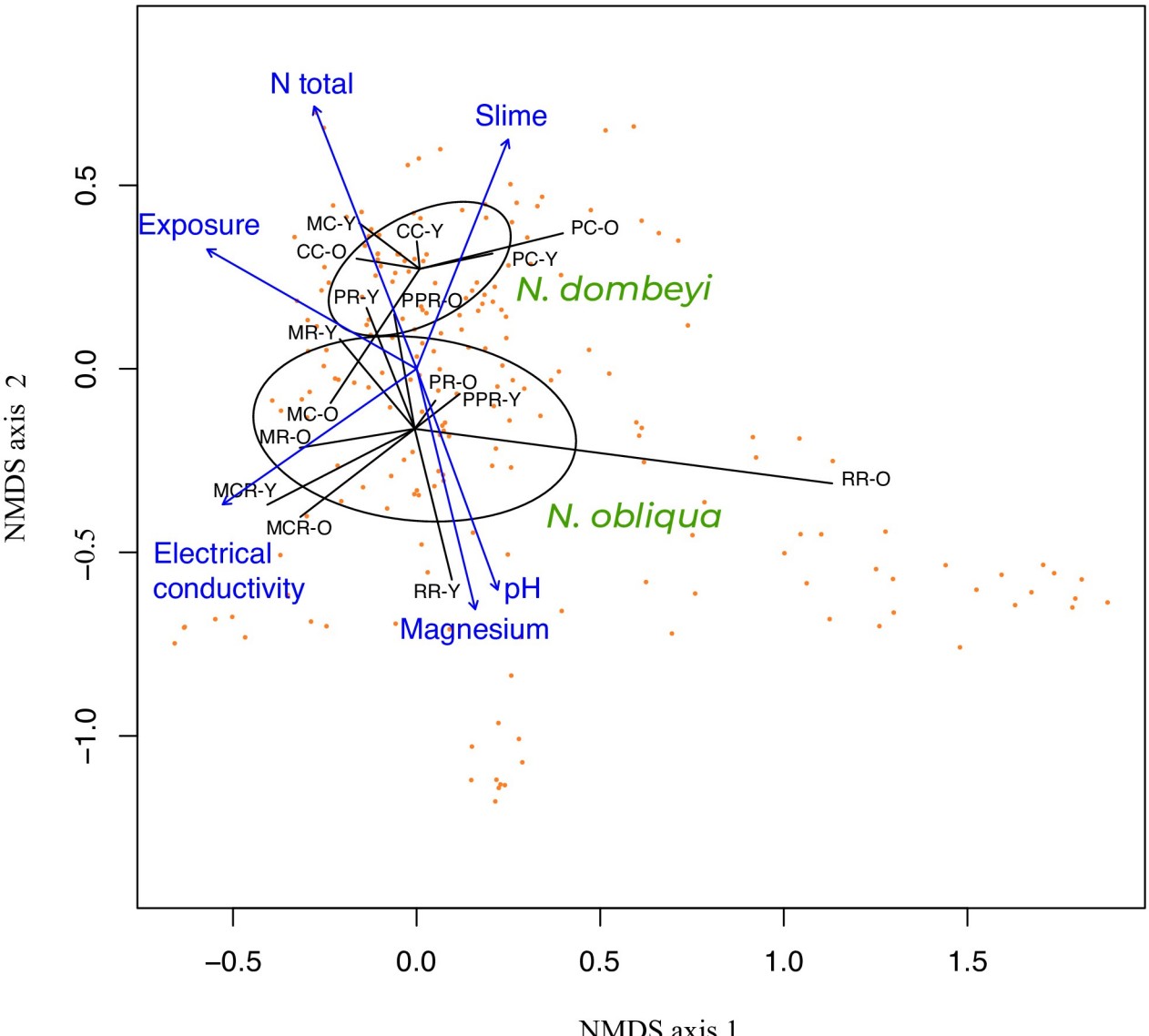

**Fig 4. NMDS plot of fungal communities associated with root tips (ECMs) and the surrounding soil from eight monospecific and mixed *Nothofagus dombeyi* and *N. obliqua* stands based on the ITS2 sequences.** For the site abbreviations, see Fig 1. Orange dots represent individual fungal genera with a minimum relative abundance of 0.1%. Environmental variables with $p < 0.01$ are displayed as blue arrows. Samples are grouped according to the corresponding tree species and marked as coming either from old (O) or young (Y) trees.

in boreal and temperate forests of the Northern Hemisphere (e.g., [28]). Although only the ECM tips were sampled, the approach allowed us to sequence other fungal groups with functions ranging from saprotrophic to pathogenic (see Fig 5).

Similar to the study by Nouhra et al. [8], we observed no statistical differences in the main fungal partners associated with *N. dombeyi* and *N. obliqua*. This result suggests that both plant species practically coexist at the sampled sites, and the remaining old trees after logging served as sources of fungal inoculum for the new generation of trees [43]. Additionally, both tree species grow on soil with broadly similar characteristics, they have similar ages, and these are phylogenetically related. Although some sites showed greater anthropogenic influence on both the

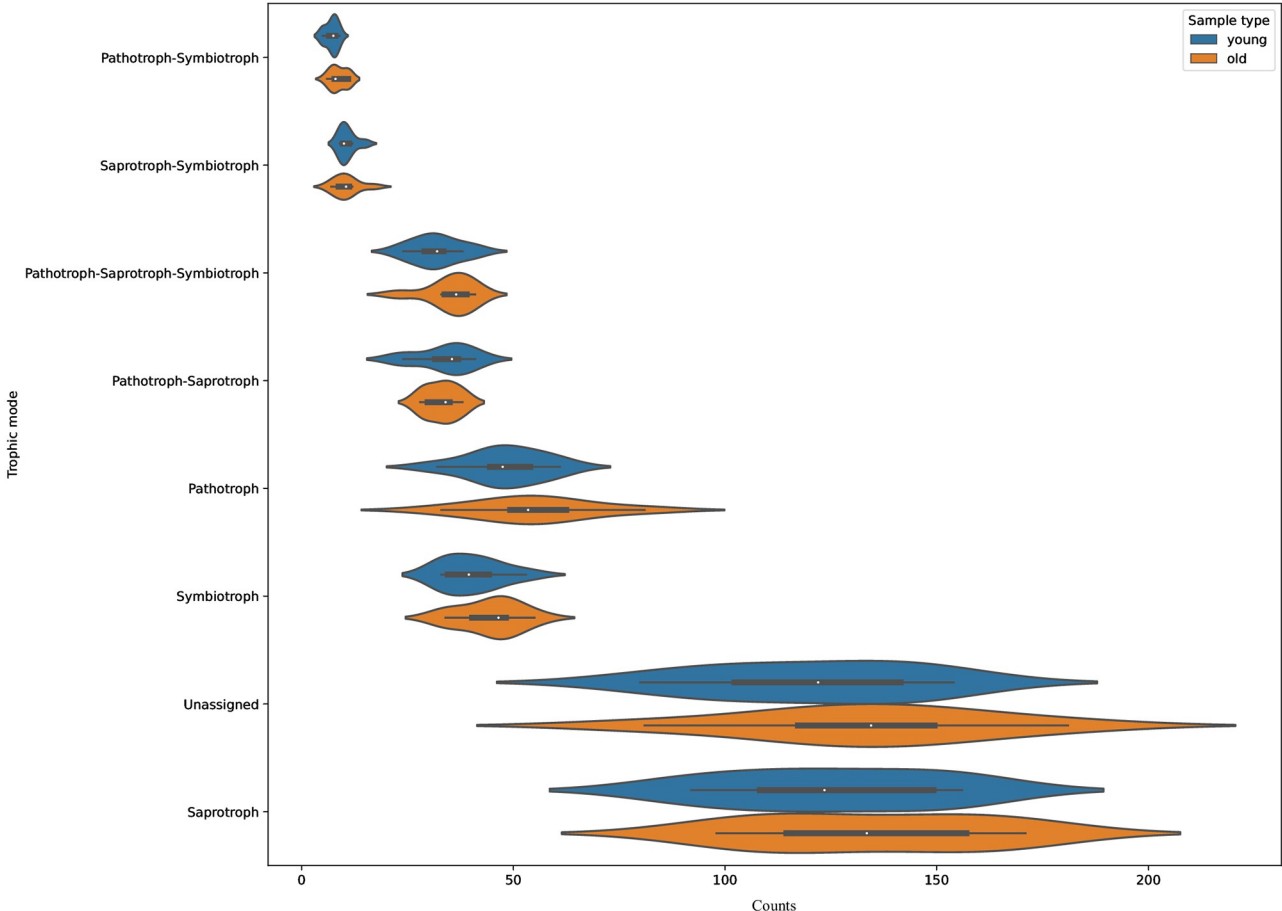

**Fig 5. Ecological associations of taxa from fungal communities in root tips and the surrounding soil from *Nothofagus dombeyi* and *N. obliqua* stands.** Samples from old and young trees are compared. Broad ecological categories were predicted with FunGuild at the genus level. White dots represent the median values in each group. The interquartile range corresponds to the black bar in the centre of each violin. The kernel density plot for each category is displayed in blue and orange, corresponding to the trees' age.

forest and soil, this does seem to significantly impact on the diversity of the belowground fungal communities (see Fig 4).

## Discrepancies between fruiting body occurrences and ectomycorrhizal root tip sequencing

Our results revealed the high diversity of fungi compared with the limited number of fungal species represented by fruiting bodies that could be observed in the field. Several previous studies, both in the Northern Hemisphere and in South America, indicated that fungal diversity studies based on DNA sequencing allow the detection of fungal groups without conspicuous fruiting bodies. For example, Atheliales, Thelephorales, Sebacinales (Basidiomycota), Helotiales, and Pezizales (both Ascomycota) either do not produce any fruiting bodies or develop inconspicuous fruiting bodies and are therefore often not detected during field sampling. It should be noted that the fungal groups detected to be dominant in our study are the same as those that dominate in boreal and temperate forests in the Northern Hemisphere (e.g., [28]). Our Illumina sequencing analysis showed the absence or low relative abundance of *B.*

*loyo* and *Ramaria* spp. at sites where they are abundant in terms of fruiting bodies, which could be caused by sampling bias, specifically because the roots were taken very close to the trunk and at shallow depths. In contrast, both species grow in circles (fairy rings) around the trees, which could not be sampled because of deep root growth below the fruiting bodies. Although Lian et al. [44] found that the ECM communities within and outside of the fairy rings were similar, there are also other cases where it has been observed that the aboveground and belowground occurrences of ECM species do not overlap (e.g., [45]).

## Impact of using Illumina sequencing for sustainable forest management decisions

Despite the enormous effort put into monitoring ecosystems via traditional methods based on visual identification and cultivation, these provide an incomplete assessment of the diversity that exists within the ecosystems [46, 47]. This issue is a major impediment in studying micro-organisms such as fungi, which do not all form fructifications, are practically invisible to the human eye, or are so ephemeral that they easily escape field sampling. In addition, most fungi are not cultivable under laboratory conditions, as they establish obligate parasitic or symbiotic associations with plants. Consequently, monitoring biodiversity and ecosystem functionality through DNA sequencing offers advantages over traditional methods, as it allows a more complete diversity to be surveyed and allows fungi to be detected from their vegetative structure (hyphae or mycelium), regardless of whether they produce fruiting bodies or not. It should be noted that the production of fruiting bodies involves various environmental and biological factors specific to each species, meaning that some fungi may fructify very sporadically, including *B. loyo* and *Ramaria* spp. [48]. More accurate taxonomic identification (DNA barcoding) of fungal species can also be achieved, and thus more detailed tracking of particular species. In conclusion, Illumina sequencing is a useful tool for detecting changes in particular taxa or wider biodiversity, opening the possibility of being able to assess the ecological status of a given ecosystem through such metrics. The potential benefits of Illumina sequencing of fungal communities for fungal monitoring programmes include characterization of their food webs, assessment of responses to their disruption and stress, and the detection of sensitive, rare, invasive, edible, and toxic species, etc.

To conclude, our study represents a first glance at the fungal communities hosted by young and old *N. obliqua* and *N. dombeyi* trees using environmental ITS2 sequencing in an area considered a biodiversity hotspot of southern Chile. As well as the characterization of ECM fungi, our results provide a view of the greater diversity of fungi, including previously undetected species (i.e., not detected as fruiting bodies), despite the imbalance in the forest stands sampled. These findings revealed a high diversity and non-significant compositional differences in the belowground fungal communities of *N. obliqua* and *N. dombeyi* trees, suggesting the impact of recolonization, similar anthropogenic use, soil conditions, and proximity between the sampled sites.

The findings of this study show that Illumina sequencing is a powerful technique in molecular ecology that allows for the simultaneous identification of multiple species from environmental DNA samples. Therefore, this method has immense potential in various fields, including forest rehabilitation, restoration, and forest management practices. Finally, our study further highlights an attempt to link below- and aboveground occurrences of the edible mushrooms *B. loyo* and *Ramaria* spp. As we detected great variations in the abundance of fruiting bodies vs. the abundance in the rhizosphere, these differences must be considered in the future to fully understand the distribution of these taxa. With this study, we hope to offer a starting point for future research in this area.

## Supporting information

**S1 Table.**
(XLSX)

**S2 Table.**
(XLSX)

**S3 Table.**
(XLSX)

**S4 Table.**
(XLSX)

**S5 Table.**
(XLSX)

**S6 Table.**
(XLSX)

**S1 Fig.**
(PDF)

**S2 Fig.**
(PDF)

**S3 Fig.**
(PDF)

## Acknowledgments

We thank C. Stuardo, I. Montenegro, V. Claramunt, and E. Molina for collecting the material and for providing pictures of the collection sites and fruiting bodies. Finally, we thank J. Palma and E. Molina for inviting us to be part of and contribute to the project. MES thanks M.G. Fischer and I. Schlichting for their support.

## Author Contributions

**Conceptualization:** Sigisfredo Garnica.

**Data curation:** Max Emil Schön.

**Funding acquisition:** Max Emil Schön, Sigisfredo Garnica.

**Investigation:** Sigisfredo Garnica.

**Methodology:** Max Emil Schön, Sarah Zuern, Milena Vera.

**Project administration:** Sigisfredo Garnica.

**Resources:** Sigisfredo Garnica.

**Supervision:** Sigisfredo Garnica.

**Validation:** Sigisfredo Garnica.

**Visualization:** Max Emil Schön.

**Writing – original draft:** Max Emil Schön, Sigisfredo Garnica.

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
