## [Decision Letter · Decision Letter 0]

15 Jun 2023

PONE-D-23-14137Metabarcoding of root-associated ectomycorrhizal fungal communities in andean-patagonian *Nothofagus* forestsPLOS ONE

Dear Dr. Garnica,

Thank you for submitting your manuscript to PLOS ONE. After careful consideration, we feel that it has merit but does not fully meet PLOS ONE’s publication criteria as it currently stands. Therefore, we invite you to submit a revised version of the manuscript that addresses the points raised during the review process.

Please submit your revised manuscript by Jul 30 2023 11:59PM. If you will need more time than this to complete your revisions, please reply to this message or contact the journal office at plosone@plos.org. Please include the following items when submitting your revised manuscript:A rebuttal letter that responds to each point raised by the academic editor and reviewer(s). You should upload this letter as a separate file labeled 'Response to Reviewers'.A marked-up copy of your manuscript that highlights changes made to the original version. You should upload this as a separate file labeled 'Revised Manuscript with Track Changes'.An unmarked version of your revised paper without tracked changes. You should upload this as a separate file labeled 'Manuscript'.If applicable, we recommend that you deposit your laboratory protocols in protocols.io to enhance the reproducibility of your results. Protocols.io assigns your protocol its own identifier (DOI) so that it can be cited independently in the future. For instructions see: https://journals.plos.org/plosone/s/submission-guidelines#loc-laboratory-protocols. Additionally, PLOS ONE offers an option for publishing peer-reviewed Lab Protocol articles, which describe protocols hosted on protocols.io. Read more information on sharing protocols at https://plos.org/protocols?utm_medium=editorial-email&utm_source=authorletters&utm_campaign=protocols.

We look forward to receiving your revised manuscript.

Kind regards,

Alejandro Huertas Herrera

Academic Editor

PLOS ONE

Journal Requirements:

"We thank C. Stuardo, I. Montenegro, V. Claramunt and E. Molina for carried out the collection of material and by providing pictures of collecting sites and fruiting bodies. We thank all the researchers involved in the project FYT-2018-0723. Finally, we thank J. Palma and E. Molina for inviting us to be part of and contribute to the project. MES would like to thank M.G. Fischer for support."

"SG: The field and laboratory research were financially supported by the Fundación para la Innovación Agraria (FIA) belonging to the Ministry of Agriculture of the Government of Chile (Grant FYT-2018-0723). 

MES: This work was supported by the Max Planck Society.

5. Please note that in order to use the direct billing option the corresponding author must be affiliated with the chosen institute. Please either amend your manuscript to change the affiliation or corresponding author, or email us at plosone@plos.org with a request to remove this option.

8. Please upload a copy of Figure 6, to which you refer in your text on page 25. If the figure is no longer to be included as part of the submission please remove all reference to it within the text.

9. We note that Figure 1 in your submission contain [map/satellite] images which may be copyrighted. All PLOS content is published under the Creative Commons Attribution License (CC BY 4.0), which means that the manuscript, images, and Supporting Information files will be freely available online, and any third party is permitted to access, download, copy, distribute, and use these materials in any way, even commercially, with proper attribution. For these reasons, we cannot publish previously copyrighted maps or satellite images created using proprietary data, such as Google software (Google Maps, Street View, and Earth). For more information, see our copyright guidelines: http://journals.plos.org/plosone/s/licenses-and-copyright.

Additional Editor Comments:

Dear authors,

I am writing to provide you with the feedback received from the reviewers regarding your manuscript, which you submitted for consideration in PlosOne. I have received two of three pieces of feedback from the reviewers who agreed to review your manuscript. I am pleased to inform you that both reviewers have expressed their support for the publication of your work, a decision that I share as well. However, several minor revisions need to be addressed to proceed with the publication process.

Firstly, I would like to draw your attention to the comments raised by the reviewers. I kindly request that you incorporate explanations in the methods or discussion sections, clarifying the reasons behind the limited data points collected. This will provide further context and enhance the overall understanding of your research.

Furthermore, I would like to emphasize the significance of the formatting corrections pointed out by the reviewers, which were attached in a PDF file. These adjustments are crucial for ensuring the clarity of your manuscript. 

Best regards,

Reviewers' comments:

Reviewer's Responses to Questions

**Comments to the Author**

1. Is the manuscript technically sound, and do the data support the conclusions?

Reviewer #1: Yes

Reviewer #2: Yes

2. Has the statistical analysis been performed appropriately and rigorously? 

Reviewer #1: Yes

Reviewer #2: Yes

3. Have the authors made all data underlying the findings in their manuscript fully available?

Reviewer #1: Yes

Reviewer #2: Yes

4. Is the manuscript presented in an intelligible fashion and written in standard English?

Reviewer #1: Yes

Reviewer #2: Yes

5. Review Comments to the Author

Reviewer #1: The paper "Metabarcoding of root-associated ectomycorrhizal fungal communities in andean-pata gonian Nothofagus forests" describes the ectomycorrhizal microbiomes of Southern American typical woodlands. This is an important piece of work, since Nothofagus ectomycorrhiza are underresearched.

The title startiing with "metabarcoding" is a bit nondescript. Also, when I read metabarcoding, I am usually not interested anymore because I expect some boring data assembly without a real in-depth analysis. In addition, the title states "root-associated ectomycorrhizal", which is an unnecessary repetition. A more descriptive title would add interest, maybe something along the lines "From fungal saprobionts to symbiont ectomycorrhiza on Nothofagus"?

The abstract with over 300 words is longish. May parts are not necessary and should be deleted, including which region was sequenced (which actually indicates that this is a mycobiome study, not a metabarcoding, since no barcodes are focussed upon...), or alluding to future testing of fairy rings (which, by the way, do not necessarily sit above the mycorrhizal root tips, as long exploration type mycorrhiza would have ample soil mycelium far away from the tree...).

The subheadings in Results are rather derived from the methods used. This is not paying the interesting results justice. It would be much wiser to find more descriptive titles like "Basidiomycota dominated the mycobiome in Nothofagus forests" or so. Especially the first two sub-chapters should be re-configured in this way.

Fig. 3 could go to supplemental material, as the presentation does not show obvious differences. Rather, it might be wise to include the table on soil analyses and add a paragraph on soil differences with relation to evergreen vs. deciduous and monospecific vs. mixed forest on soil properties before going into detail.

Also, in my opinion the species composition might be worth a bit more representation. Maybe split the fungal classes to give Murorales, Asco- and Basidiomycetes their own representation on order level? Here, suppl. Tab. S5 needs to go into the main manuscript, as it contains important information.

Should "electric charge" be electrical conductivity?

In discussion, the sub-paragraph "Discrepancies between fruiting body occurrences and ecto 381 mycorrhizal root tip sequencing: the cases of Boletus loyo 382 and Ramaria spp." is not astonishing or does not contain new information, as this fact has been often recognized. I would not make a special point out of it, maybe a very short sentence in one of the other sub-chapters is sufficient. Especially, since your approach was definitely not a thorough sampling campaign for fruitbodies.

Similarly, the sub-heading "Impact of using metabarcoding for sustainable forest man 403 agement decision" could be shortened. The sub-chapter is important for application, but maybe a shorter version may be sufficient.

The supplemental material consists of excel sheets without the necessary information to correctly read and interpret the tables. This should be amended with a written supplmental material part. (and may be included tables in a non-excel format within one pdf)

Did the authors consider uploading not only the raw sequencing results, but the mycobiome analyses for all their sites to Genbank?

All in all, the work reports very interesting and novel findings and should be published after minor revision.

Reviewer #2: Dear authors,

Your manuscript entitled " Metabarcoding of root-associated ectomycorrhizal fungal communities in andean-patagonian Nothofagus forests." provides valuable insights into the ectomycorrhizal fungal communities in native Andean-Patagonian Nothofagus forests and highlights the importance of factors such as tree age, soil properties, and human activities in shaping these fungal communities. However, the manuscript requires major revisions. I have included my comments and suggestions in the attached file.

Bests regards,

1. Is the manuscript technically sound, and do the data support the conclusions?

Yes. the manuscript describe a technically sound piece of scientific research with data that supports the conclusions.

2. Has the statistical analysis been performed appropriately and rigorously?

Yes, the statistical analyses are appropriate and adequate for the data in this study.

3. Have the authors made all data underlying the findings in their manuscript fully available?

Yes, all the findings are available in this manuscript.

Yes, the manuscript is presented in a clear and understandable manner, and it is written in standard English.

6. PLOS authors have the option to publish the peer review history of their article (what does this mean?). If published, this will include your full peer review and any attached files.

Reviewer #1: No

Reviewer #2: No

---

## [Author Response · Author response to Decision Letter 0]

29 Jul 2023

Reviewer #1: The paper "Metabarcoding of root-associated ectomycorrhizal fungal communities in andean-pata gonian Nothofagus forests" describes the ectomycorrhizal microbiomes of Southern American typical woodlands. This is an important piece of work, since Nothofagus ectomycorrhiza are underresearched.

The title startiing with "metabarcoding" is a bit nondescript. Also, when I read metabarcoding, I am usually not interested anymore because I expect some boring data assembly without a real in-depth analysis. In addition, the title states "root-associated ectomycorrhizal", which is an unnecessary repetition. A more descriptive title would add interest, maybe something along the lines "From fungal saprobionts to symbiont ectomycorrhiza on Nothofagus"?

We changed the title of our manuskript. The reviewer ist right. 

The abstract with over 300 words is longish. May parts are not necessary and should be deleted, including which region was sequenced (which actually indicates that this is a mycobiome study, not a metabarcoding, since no barcodes are focussed upon...), or alluding to future testing of fairy rings (which, by the way, do not necessarily sit above the mycorrhizal root tips, as long exploration type mycorrhiza would have ample soil mycelium far away from the tree...).

The reviewer ist right. Just using only ITS2 is too uninformative to be considered as a barcode. Following this principle, we have changed barcode to illumina sequencing.

The subheadings in Results are rather derived from the methods used. This is not paying the interesting results justice. It would be much wiser to find more descriptive titles like "Basidiomycota dominated the mycobiome in Nothofagus forests" or so. Especially the first two sub-chapters should be re-configured in this way.

We changed the subheadings in the results part of the manuscript as suggested.

Fig. 3 could go to supplemental material, as the presentation does not show obvious differences. Rather, it might be wise to include the table on soil analyses and add a paragraph on soil differences with relation to evergreen vs. deciduous and monospecific vs. mixed forest on soil properties before going into detail. 

We decided to follow this suggestion and moved Fig 3 to the supplemental material.

Also, in my opinion the species composition might be worth a bit more representation. Maybe split the fungal classes to give Murorales, Asco- and Basidiomycetes their own representation on order level? Here, suppl. Tab. S5 needs to go into the main manuscript, as it contains important information.

We added a new Figure, showing the relative abundances of fungal orders in a heatmap. We also provide such heatmaps for the phylum and genus level in the supplemental material.

Should "electric charge" be electrical conductivity?

Indeed, it should be. We changed the figure accordingly.

In discussion, the sub-paragraph "Discrepancies between fruiting body occurrences and ectomycorrhizal root tip sequencing: the cases of Boletus loyo and Ramaria spp." is not astonishing or does not contain new information, as this fact has been often recognized. I would not make a special point out of it, maybe a very short sentence in one of the other sub-chapters is sufficient. Especially, since your approach was definitely not a thorough sampling campaign for fruitbodies.

The reviewer is correct that as part of the methodology of this study was not included the collection of fruiting bodies. However, this part of our paper refers rather to the experience of collecting the mushrooms in question by the indigenous people of that region. Hence, we considered mentioning this aspect in our discussion.

Similarly, the sub-heading "Impact of using metabarcoding for sustainable forest management decision" could be shortened. The sub-chapter is important for application, but maybe a shorter version may be sufficient.

In the revised version of our manuscript we have focused on Illumina sequencing rather than metabarcoding taking into consideration the suggestions of one of the reviewers. We have added a paragraph regarding its application for forest management and protection purposes.

The supplemental material consists of excel sheets without the necessary information to correctly read and interpret the tables. This should be amended with a written supplmental material part. (and may be included tables in a non-excel format within one pdf)

We added a separate supplemental material file with table and figure descriptions. We include heatmaps of the tables describing fungal diversity for easier visual inspection.

Did the authors consider uploading not only the raw sequencing results, but the mycobiome analyses for all their sites to Genbank?

While we considered this suggestion carefully, we do not think it is necessary and therefore chose not to do it.

All in all, the work reports very interesting and novel findings and should be published after minor revision.

Reviewer #2: Dear authors,

Your manuscript entitled " Metabarcoding of root-associated ectomycorrhizal fungal communities in andean-patagonian Nothofagus forests." provides valuable insights into the ectomycorrhizal fungal communities in native Andean-Patagonian Nothofagus forests and highlights the importance of factors such as tree age, soil properties, and human activities in shaping these fungal communities. However, the manuscript requires major revisions. I have included my comments and suggestions in the attached file.

The changes suggested by the reviewer were included in the improved version of our manuscript. We have also included the reviewer's suggested information in Table S1.

Bests regards,

---

## [Decision Letter · Decision Letter 1]

8 Aug 2023

High diversity of fungal ecological groups in Andean–Patagonian Nothofagus forests

PONE-D-23-14137R1

Dear Dr. Garnica,

We’re pleased to inform you that your manuscript has been judged scientifically suitable for publication and will be formally accepted for publication once it meets all outstanding technical requirements.

Kind regards,

Alejandro Huertas Herrera

Academic Editor

PLOS ONE

Additional Editor Comments (optional):

Dear authors,

Thank you for considering the suggestions and changes for your paper. Both reviewers have accepted the manuscript, and so do I. However, please consider the following minor changes to improve it.

Lines 8 and 9: Change the dot for a comma in 9,600 and 1,125.

Line 17: Change major to significant.

Line 76: Delete “among others.”

Line 99: Change* Boletus loyo* to *B. loyo*.

Line 128: Change the dot to a comma in the 3,292 km2.

Line 128: Add a space between the dot and the word “The.”

Line 143: (PR) Pultro road site, (PC) Pultro road site needs to be clarified; please use the same way to present the sites as shown in line 128 (The sites at Pultro Road (PR and PC)).

Line 146: Please decide if “old” or “mature” will describe this kind of stand.

Line 149: Change ectomycorrhizal to tree.

Lines 161 and 162: The S1 Table also contains soil property data. Please inform this data availability in the sentence or another part of the manuscript, e.g., Root and soil sampling.

Lines 190, 277, 278, 285: Add a comma to 9,600, 8,813, 8,101, 1,125.

Line 209: Delete "a total of."

Line 262: Delete "etc."

Line 316: Decide if “old” or “mature” will describe this kind of stand.

Line 343: The dots look orange. Please change the word red to orange.

Line 343: Although the figure is well-intuitively explained, please indicate or clarify what the black lines and ellipses correspond to.

Line 350: Please delete "of”.

Lines 350, 356, 345, and 433: Please decide if “old” or “mature” will describe this kind of stand.

Line 355: Delete "major."

Line 361: Please decide if “old” or “mature” will describe this kind of stand. This change must be reflected in Figure 5.

Line 371: *N. obliqua* in italics.

Lines 388 and 389: Please delete “: the cases of *Boletus loyo* and* Ramaria* spp.”

Line 431: Delete 431.

I understand that changing the figure font can be tedious, but as a rule, the manuscript must present the same font (e.g., Arial or Times New Roman). Please keep just one font style throughout the manuscript to avoid future publishing process issues in PlosOne.

Best regards,

Alejandro

Reviewers' comments:

Reviewer's Responses to Questions

**Comments to the Author**

1. If the authors have adequately addressed your comments raised in a previous round of review and you feel that this manuscript is now acceptable for publication, you may indicate that here to bypass the “Comments to the Author” section, enter your conflict of interest statement in the “Confidential to Editor” section, and submit your "Accept" recommendation.

Reviewer #1: All comments have been addressed

Reviewer #2: All comments have been addressed

2. Is the manuscript technically sound, and do the data support the conclusions?

Reviewer #1: Yes

Reviewer #2: Yes

3. Has the statistical analysis been performed appropriately and rigorously? 

Reviewer #1: Yes

Reviewer #2: Yes

4. Have the authors made all data underlying the findings in their manuscript fully available?

Reviewer #1: Yes

Reviewer #2: Yes

5. Is the manuscript presented in an intelligible fashion and written in standard English?

Reviewer #1: Yes

Reviewer #2: Yes

6. Review Comments to the Author

Reviewer #1: The authors did a Gold Job in addressing all queries. All questions werde aapropriately answered and the manuscript has been improved. .

Reviewer #2: (No Response)

7. PLOS authors have the option to publish the peer review history of their article (what does this mean?). If published, this will include your full peer review and any attached files.

Reviewer #1: No

Reviewer #2: No

---

## [Editor Report · Acceptance letter]

15 Aug 2023

PONE-D-23-14137R1 

High diversity of fungal ecological groups in Andean–Patagonian *Nothofagus* forests 

Dear Dr. Garnica:

I'm pleased to inform you that your manuscript has been deemed suitable for publication in PLOS ONE. Congratulations! Your manuscript is now with our production department. 

Kind regards, 

on behalf of

Dr. Alejandro Huertas Herrera 

Academic Editor

PLOS ONE